# Cracking of Gem Opals

**Boris Chauviré** [1,*], **Valentin Mollé** [2], **Florine Guichard** [3], **Benjamin Rondeau** [4], **Paul Stephen Thomas** [5] **and Emmanuel Fritsch** [6]

1   GeoGems, 44500 Guérande, France
2   Université d'Orléans, CNRS, BRGM, ISTO, UMR 7327, 45071 Orléans, France
3   Faculté des Sciences et Technologies, Boulevard des Aiguillettes, 54506 Vandœuvre-lès-Nancy, France
4   Laboratoire de Planétologie et Géosciences, Nantes Université, Univ Angers, Le Mans Université, CNRS, UMR 6112, 44000 Nantes, France
5   School of Civil and Environmental Engineering, University of Technology, Sydney, NSW 2007, Australia
6   Institut des Matériaux de Nantes Jean Rouxel, Nantes Université, CNRS, IMN, 44000 Nantes, France
*   Correspondence: boris.chauvire@geogems.fr

**Abstract:** The value of gem opals is compromised by their potential susceptibility to "crazing", a phenomenon observed either in the form of whitening or cracking. To understand the latter, 26 opal samples were investigated and separated into 2 groups based on handling: "water-stored" opal samples, which are stored in water after extraction, and "air-stored" opal samples, which are stored in air for more than a year. To induce cracking, samples were thermally treated by staged heating and characterized using optical microscopy and Raman spectroscopy before and after cracking. For water-stored opals, cracking was initiated with moderate heating up to 150 °C, while for air-stored opals, higher temperatures, circa 300 °C, were required. In water-stored opals that cracked, polarized light microscopy revealed stress fields remaining around the cracks, and a red shift in the Raman bands suggested tensile stresses. These stresses were not observed in air-stored samples that cracked. Based on these observations, for air-stored samples, cracking was ascribed to super-heated water-induced decrepitation. By contrast, for water-stored samples, cracking was linked to drying shrinkage, which correlates with the anecdotal reports from the gem trade. We thus identify the physical origin of cracking, and by comparing it to current knowledge, we determine the factors leading to cracking.

**Keywords:** opal; cracking; water; TGA; drying; shrinkage; decrepitation

## 1. Introduction

Opals are well-known gems offering a wide range of aspects and optical phenomena, particularly light diffraction in "noble" opals. The very best specimens reach very high prices in the market. However, the reputation of gem opals is tarnished by its potential to destabilize (or "craze") in two different ways: It may lose its transparency ("whitening") or develop fissures (cracking) either at the surface or in its center (examples of cracked opals are shown in Figure 1) [1]. This, of course, alters considerably the value of gem opals. Both phenomena have already been described, but they are not fully understood. Early studies have shown that "whitening" is correlated with an increase in porosity and a reorganization of the hydration states without changing the silica framework [2,3]. The first empirical investigations on cracking suggest that the drying of opals is a key factor for understanding cracking [4–6]. Nevertheless, there remains a need for an experimental or theoretical framework to recognize ahead of time that opals may "craze". This paper concentrates on the understanding of crazing due to cracking. In order to address the issue of instability in opals and its detection, the materials and their structure and properties must first be understood. Therefore, we first review current knowledge on opal structure, water content, and stability and then experimentally crack a variety of opals to assess the physical origin of this degradation process.

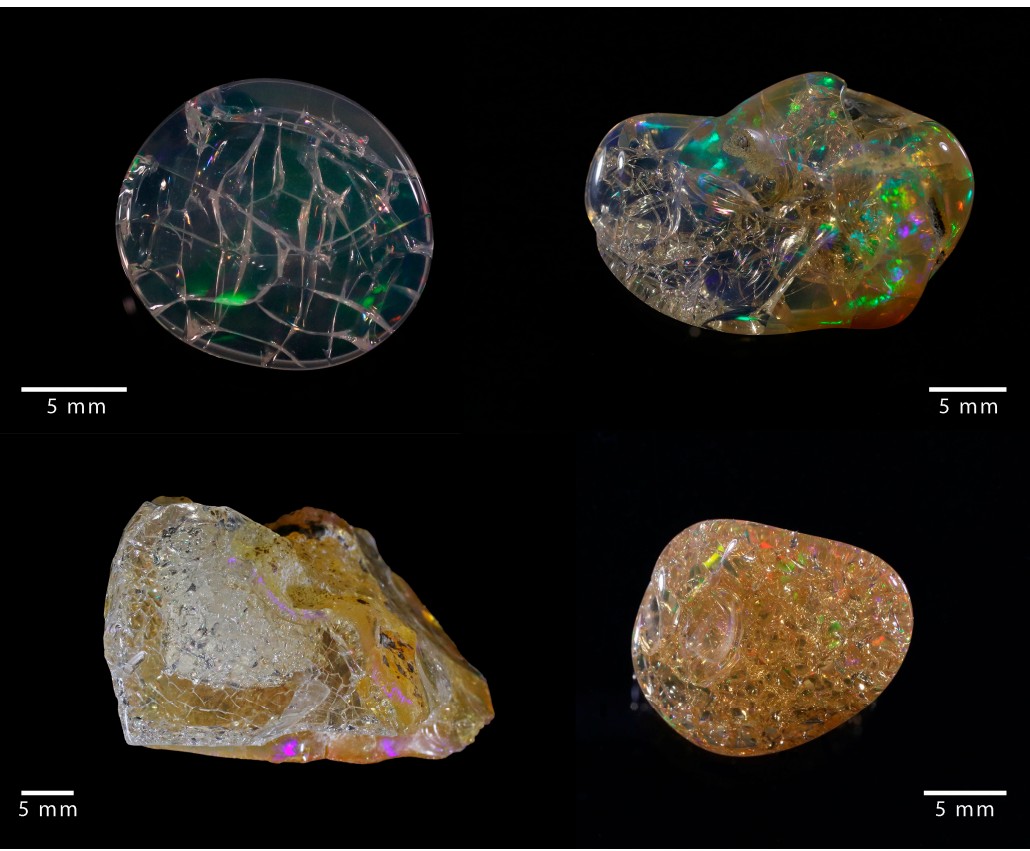

**Figure 1.** Examples of cracked opal, where a network of fractures developed.

### 1.1. What Is Opal?

Opal refers to natural hydrated varieties of non- to nanocrystalline silica ($SiO_2 \cdot nH_2O$) frequently found in a wide range of geological contexts where the aqueous alteration of silicate minerals occurs. A crystallographic classification based on X-ray diffraction (XRD) distinguishes amorphous opal (opal-A) from paracrystalline opal. The latter is further subdivided into opal-CT, where broad cristobalite and tridymite diffraction peaks are observed, and opal-C, where only cristobalite peaks are visible [7]. It has been suggested that opal-CT is a poorly crystallized cristobalite with tridymite stacking [7], but recent structural studies indicated that opal-CT may comprise the nano-domains of diffracting tridymites alternating with non-diffracting silica, with no evidence of cristobalite domains [8–11]. Opal-A, displaying a broad band typical of amorphous material in XRD, can be subdivided into two sub-types [12]:

- A network similar to amorphous silica (opal-$A_N$ or "hyalite") with a glass-like structure, formed by the quenching of hot silica-rich solutions on cooler surfaces [13,14];
- Gel-like amorphous silica (opal-$A_G$), exhibiting a structure composed of spheres, precipitated from an aqueous solution, similarly to laboratory-grown silica-gel and well documented in Australian specimens [12,15–21].

The distinction between opal types, even if firstly defined by X-ray diffraction, can also be achieved by Raman spectroscopy [22–24], infrared spectroscopy (either in the mid-infrared sensitive to silica framework vibrations [24–26] or in the near-infrared sensitive to the hydration state [12,27,28]), nuclear magnetic resonance [29,30], and by observing microstructural features by electron microscopy [9,18,31].

Both opals types are found in various geological contexts. Opal-A is a common phase constituting the silica deposits around hot springs [32–34] or during the weathering of rocks [35,36], with a formation temperature ranging from room temperature up to 100 °C and probably more for opal-$A_N$. Similarly, opal-CT can form in a hydrothermal context

such as Mexican deposits [37] or by the weathering of volcanic rocks as observed in the Wollo province in Ethiopia [38–40] with a formation temperature ranging from ambient up to 160 °C.

Microstructural features observed in opals are all based on nanograins of ca. 25 nm for opal-A [18,41] and of 10 to 50 nm for opal-CT (with an average near 25 nm, including tridymite nano-domains) [18,42,43]. In opal-A, nanograins usually aggregate in spheres ranging from 80 nm to 8 μm in diameter. Opal-CT displays a greater variety of microstructural features where nanograins may accumulate randomly in fibers, platelets, or lepispheres [18,42]. On rare occasions, monodispersed hundred-nanometer-sized silica spheres for opal-A or lepispheres for opal-CT may be arranged into ordered arrays enabling the diffraction of visible light, producing the play-of-colour (POC, patches of moving pure spectral colors) that is highly prized in precious or noble opals [21,44].

### 1.2. Water in Opals

In opals, water is present as molecular water ($H_2O$) and chemically bound water in the form of silanol groups (Si-OH) [12,27,45–47]. Near-infrared spectroscopy indicates that much of an opal's hydration consists of molecular water [12,27,48]. Early studies suggested that hydration is concentrated in the interstices between primary spheres [19], but further developments have identified more states in which water is present: interstitial (voids between primary spheres), pore, adsorbed at the surface, and as molecular water trapped in silica cages [12,49,50]. Similarly, silanol groups appear to be located at silica interfaces (such as defects such as broken Si-O-Si bridges in the bulk silica) [12].

By collating 204 measurements (mostly by thermogravimetric analysis) from the literature [3,6,12,46,51–61] (data are available in Supplementary Materials), we observe that opal water content ranges from 0.5 to 18.1 weight percent, with an average at 6.99 wt% and a standard deviation of 3.05 (Figure 2). A very slight difference between opal-A and opal-CT (6.56 ± 2.41 wt% and 7.32 ± 3.43 wt%, respectively) is shown here between 82 opal-A and 118 opal-CT samples (four opals are not identified). Considering the average and standard deviation, statistically, opal-A and opal-CT are hardly distinguishable based on water content alone. Both opal types have a significant distribution in water content, especially at the lower water content end of the distribution. The opal types do, however, differ in terms of the shape of their distributions: Opal-A shows a distribution centred around 7 wt%, whereas opal-CT has two populations, with one close to opal-A and a second around 9 wt% (Figure 2). In addition, the difference between opals from various origins is statistically indistinguishable: Australia (70 data; 7.31 ± 2.11 wt%), Ethiopia (23 data; 8.43 ± 3.63 wt%), and Mexico (37 data; 8.32 ± 3.54 wt%).

Differential scanning calorimetry (DSC) has demonstrated that between 10 and 33% of the total water present is crystallisable [50,58]. From the melt temperature depression in DSC measurements, it is possible to estimate the pore size in which the crystallisable water is contained. It is estimated to be in the nanometer range (circa 4 to 10 nm in diameter for opal-CT and 6 to 200 (+) nm for opal-A) [50,58]. The crystallisable water contained in opal is present in both isolated and interconnected closed pores or, in some cases, in open pores exposed to the atmosphere, as demonstrated by measurements of the opal's near-infrared signature at low pressures [62]. In this study, most opals analysed have closed porosity, but some opal-CT specimens have open pores and can reversibly lose water to the atmosphere.

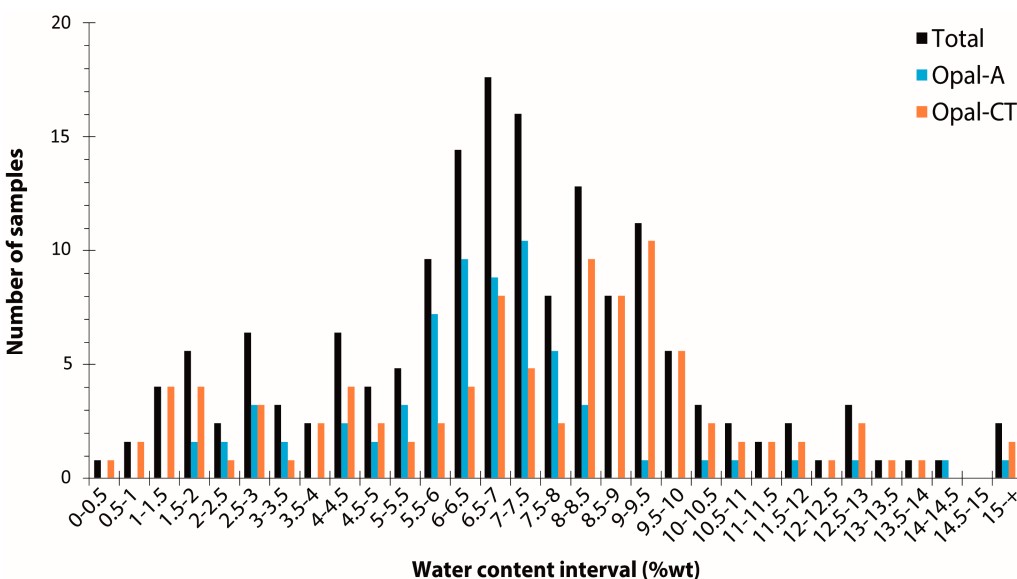

**Figure 2.** Meta-analysis of the water content found in the literature [3,6,12,46,51–61].

### 1.3. Instability of Opals by Cracking

In gemmology, instability is defined as an alteration of the aspect hindering the use of the stone for jewellery. Cracking is the development of cracks (fractures) in an initially homogeneous volume, and it obviously affects the integrity of the stone (Figure 3). In an attempt to understand cracking, early studies have attempted to induce cracking empirically, especially by sacrificing stones, although these studies provided no evidence that other samples from the same location would react similarly [4–6]. These first investigations revealed that drying rates have a significant impact on the initiation of cracking, suggesting that drying shrinkage in opals could be the main driver for cracking [5,6]. Cracks have been reported to occur primarily in transparent opals, specifically at the surface of the stone [1]. Some attempts have been made to prevent cracking in opals, particularly by applying a specific treatment after mining [63]. Despite a better knowledge of this process, no scientific nor objective criterion has been established at this point to assess or predict the stability of a given sample.

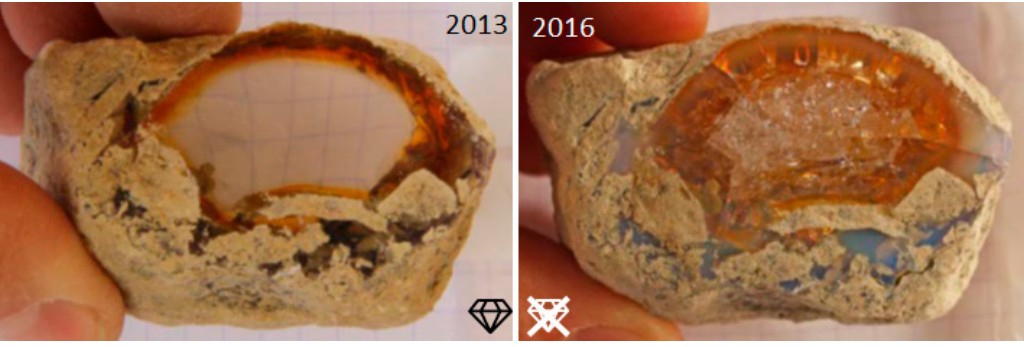

**Figure 3.** Example of an opal sample collected in 2013 at the mine site (Kok Woha, Wollo Province, Ethiopia, see [39] for details on mines) and the result after 3 years of storage in the atmosphere, where it was allowed to dry.

## 2. Materials and Methods

In this study, 26 crack-free opals have been analysed: 23 specimens are opal-CT from various origins (mainly from Ethiopia and Mexico), and 3 are opal-A (all opal-A$_G$; Table 1). Most samples are part of the opal collection of Nantes University. The 7 samples noted as KOK, CH0, and AN were collected in 2016 directly from the mine site (the letters are abbreviations for Kok Woha, Chegen, and Anset, respectively, which are the opal mines in Wollo Province, Ethiopia) by Selam Dagnachew (former MSc student at the Addis Ababa University). These samples have been preserved in a plastic tube filled with water just after extraction to avoid drying before analysis. By contrast, the 19 remaining samples from the Nantes collection have spent from one to up to 20 years in the ambient atmosphere; these are hereafter categorized as "air-stored" in contrast to the "water-stored" samples preserved in water. All samples have been sliced into thin sections of 2–2.5 mm for water-stored samples (2 pieces for KOK01, KOK04, and CH01) and from 150 to 300 μm for air-stored samples. Further details about the geological context and structure of air-stored samples are outlined in Chauviré et al. [27].

**Table 1.** Samples analysed in this study. The * outlines water-stored samples collected from the mining site and preserved in water.

| Variety | Sample | Geological Origin |
|---------|--------|-------------------|
| Opal-A$_G$ | 785 | Honduras |
| | 86.2 | Kashau, Slovakia |
| | 1040 | Coober Pedy, South Australia, Australia |
| Opal-CT | 928 | Mali |
| | 1545 | Unknown |
| | 1548 | Fougère, Brittany, France |
| | 1543a | |
| | 1543b | |
| | 1552a | San Martin, Jalisco, Mexico |
| | 1552b | |
| | 1552d | |
| | 43l | Humbolt County, USA |
| | 521 | |
| | 1551 | Mezezo, Ethiopia |
| | YM12 | |
| | 208 | |
| | FT1111 | |
| | VTB | |
| | WT86 | |
| | KOK01 * | |
| | KOK03 * | |
| | KOK04 * | Wegel Tena, Ethiopia |
| | KOK05 * | |
| | CH01 * | |
| | CH0p * | |
| | AN02 * | |

As cracking is related to dehydration [3–6], samples have been heated to dry them and thus induced cracking. Air-stored samples have been subjected to heating from room temperature to 1000 °C (with steps at 150, 300, 350, 400, 450, 500, 550, 600, 700, and 850 °C) in a furnace without purging. Then, they have been cooled down to room temperature in a purged cell to prevent rehydration. The effect of each heating step was investigated by the examination of each sample using polarized light microscopy at room temperature before cracking and after each heating episode.

Thermogravimetric analyses (TGA) were performed on 19 air-stored samples, with a Setaram Setsy 16/18 instrument, at the School of Mathematical and Physical Science, University of Technology, Sydney, Australia. Pieces of each opal specimen (from 29.5 to 83.6 mg) were placed in 130 µL platinum pans (4,5 mm diameter and 8 mm height) covered by a loose platinum cap in order to avoid the loss of material during the possible cracking. Two sets of analyses were performed: On 5 samples (521, 1040, 1543b, 1548, WT86), water loss was measured from 20 to 1100 °C with a heating rate of 2 °C min$^{-1}$ on 3 pieces of the same sample; for the 14 remaining samples, measurements were carried out from 20 to 1000 °C with a heating rate of 10 °C min$^{-1}$ on a single piece. Temperature calibration was carried out using the melting points of high-purity zinc, lead, aluminum, and copper and gold. A baseline calibration was carried out using the experimental conditions with empty crucibles. The baseline was subtracted point to point from the experimental data. The results of these analyses have already been presented in [58], and they are re-used here in light of stability processes.

Ten samples (1548, 1552a, 1543b, and the 7 samples preserved in water) have been analyzed by Raman spectroscopy by using a LabRam HR Evolution (Horiba Scientific) equipped with a 532 nm Ar-ion laser for excitation with a power between 50 and 200 mW depending on the quality of the spectra and what the sample could handle. All Raman analyses were performed in 2017 for water-stored samples and 2017 and 2018 for air-stored samples. Each spectrum was acquired with an integration time of between 25 and 60 s and a spectral resolution of 0.8 cm$^{-1}$ (for 1548, 1552a, 1543b, KOK03, and CH0p) and 3 cm$^{-1}$ for the 5 remaining water-stored samples. Spectra were acquired at room temperatures, 150 °C to 300 °C using a Linkam heating stage adapted to the Raman spectrometer.

## 3. Results

### 3.1. Visual Examinations

All samples were inspected before and after cracking. With the exception of KOK01 and An02, which were not observed to crack, water-stored samples (those preserved in water after extraction) usually cracked a few hours after exposure to air or after slight heating to 150 °C. Among the air-stored samples, eight cracked at 300 °C (1040, 1548, 1543a, 1543b, 1552a, 1552b, 1552d, and 521), and others did not show cracking during the heating steps or even at high temperatures. Additionally, we note that some samples decreased in transparency with temperature (785 and 86.2 at 300 °C, 208 at 350 °C, 1552b and 1552d at 500 °C, and progressively from 500 to 1000 °C for 1551, YM12, and WT86). Four samples (928, 1545, FT1111, and VTB) did not show any changes (neither cracking nor whitening).

Before cracking, all samples presented isotropic behavior between crossed polarizers, as expected. After cracking, observations at room temperature differed between air-stored and water-stored samples. The air-stored samples remained isotropic after cracking, whereas water-stored samples displayed clear anisotropic features around cracks (Figure 4).

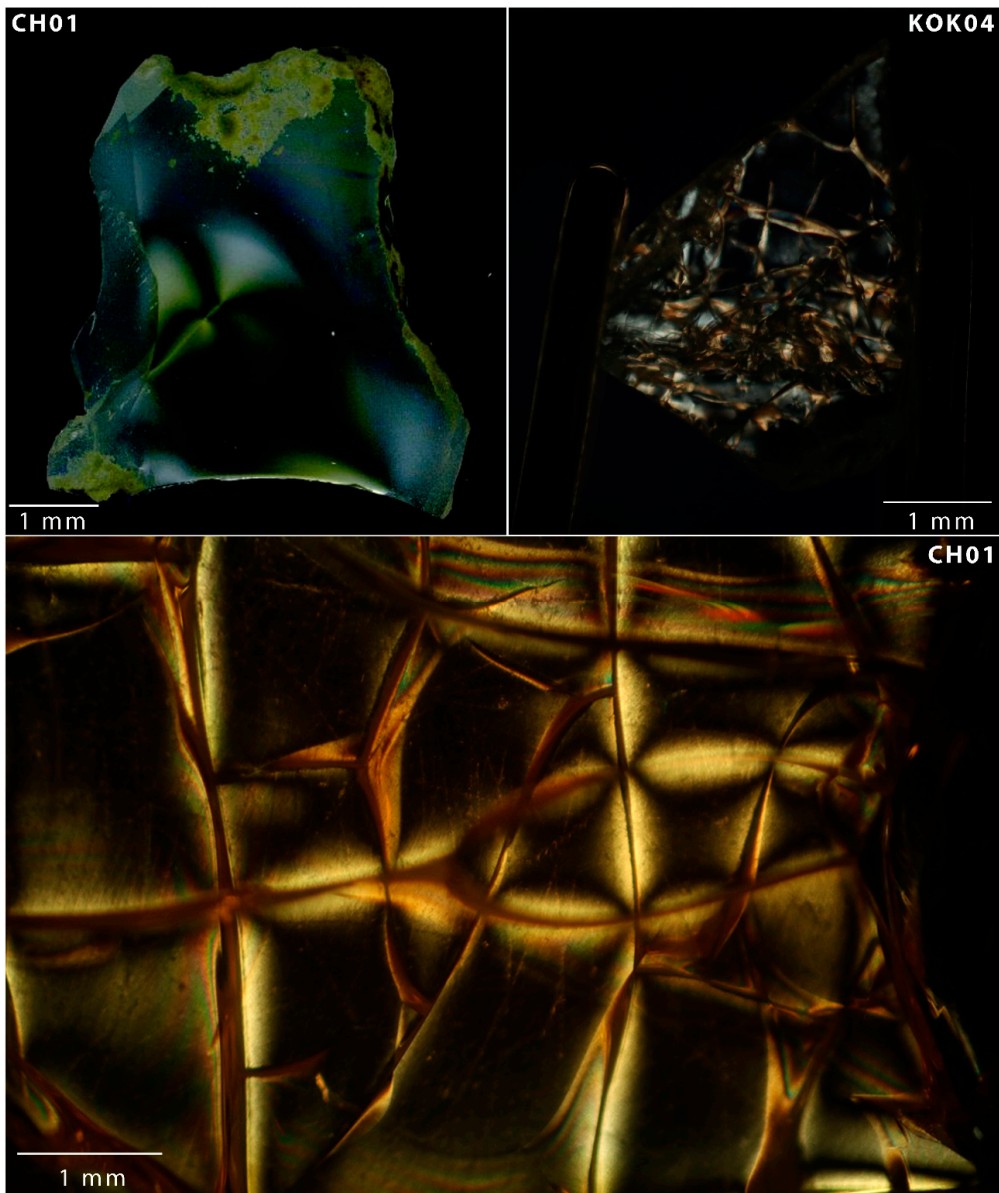

**Figure 4.** Microphotographs of samples of the two pieces of CH01 and one piece of KOK04 under cross-polarized light, with visible anisotropy around fractures.

### 3.2. Thermogravimetric Analysis (Air-Stored Samples)

Thermogravimetric analysis was not used here for the measurements of the total water content but to assess the dehydration behavior of samples. As water-stored samples are soaked in water, all samples would exhibit a first loss of water at low temperatures, interfering with the interpretation of the opal's behavior in an ambient condition. Note that the same dataset was used in Chauviré et al. [58], and the water content is presented there. Three main dehydration behaviors were observed here (Figure 5):

- No loss of water until 200–400 °C, and most dehydration (up to 90%) occurred between 300 °C and 700 °C (black trace). Observed for samples 86.2, 785, 928, 1040, 1543a, 1543b, 1545, 1548, 1552a, 1552b, and 1552d;
- Loss of water starts as soon as heating starts, and most water is lost under 300 °C (red dashed trace). Observed for samples 208, FT1111, VTB, WT86, 521, YM12, and 43l.
- Progressive irregular loss of water from room temperatures up to 900 °C for sample 1551 (blue trace).

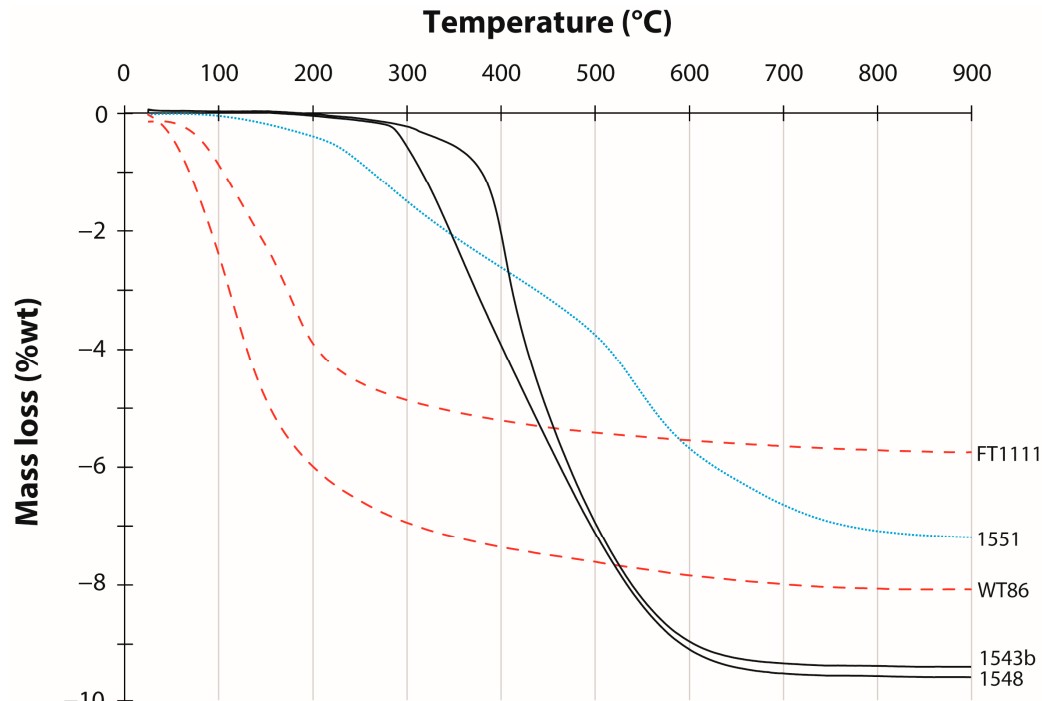

**Figure 5.** Representative thermogravimetric diagram of the three main behaviors of air-stored samples: black trace show a loss between 300 and 400 °C (sample 1543b-1548), interrupted red trace show a quick loss of water (sample FT1111-WT86), and blue trace show a progressive loss (sample 1551).

### 3.3. Raman Spectroscopy

The Raman spectra of the 10 samples analysed exhibit the typical bands of opal-CT, with the main band around 315 cm$^{-1}$, and additional bands near 780, 950 and 1070 cm$^{-1}$. Air-stored samples, at room temperature, 150 °C, or 300 °C, did not show any changes in the Raman spectra before and after cracking. By contrast, water-stored samples that cracked show a slight shift in all Raman bands from 3 to 14 cm$^{-1}$ (depending on the sample or the band considered). This shift moves toward lower frequencies after cracking (Figure 6). When cracking does not occur, the Raman spectra are identical at 25 °C and 150 °C, demonstrating that this shift is not a result of the thermal treatment.

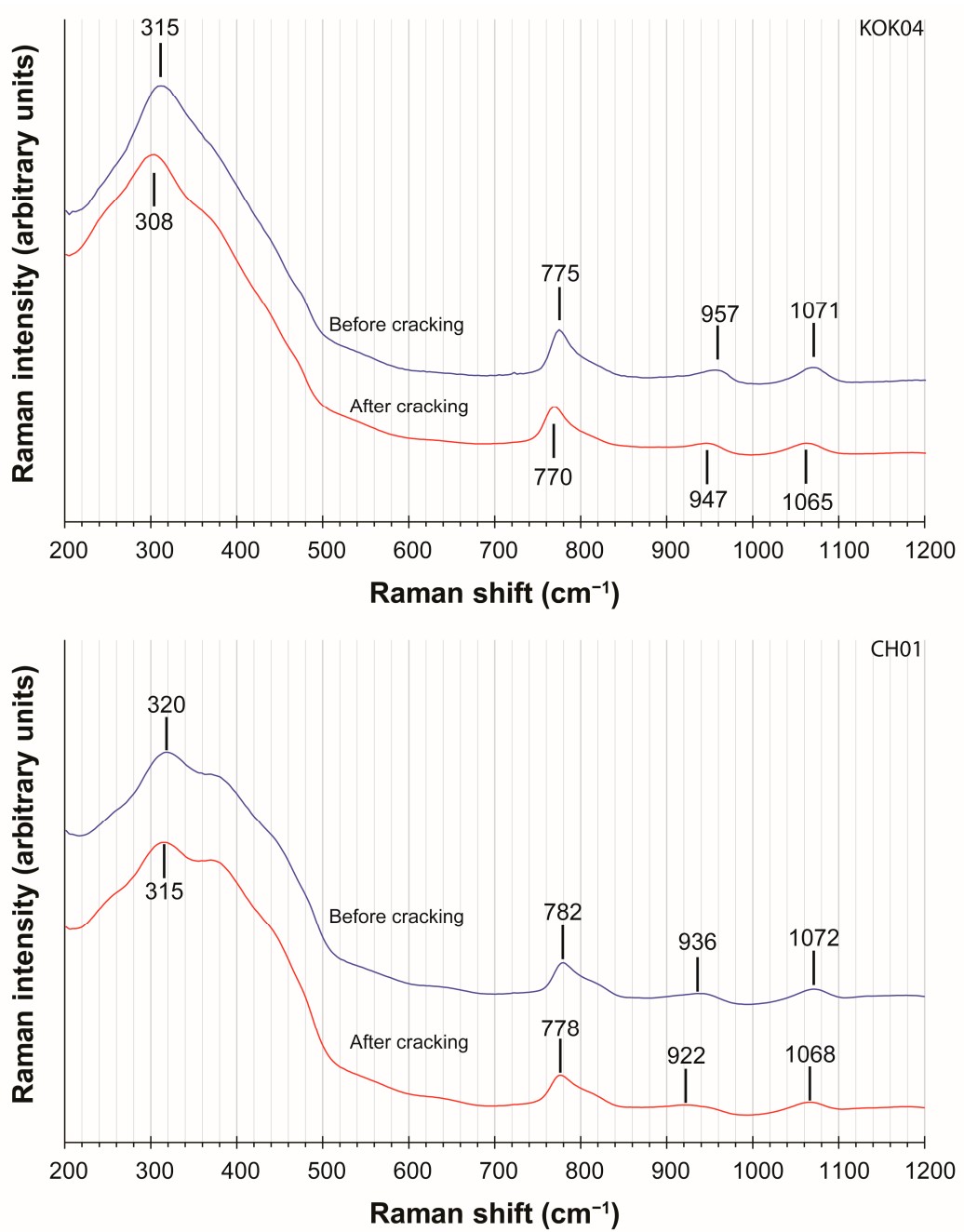

**Figure 6.** Representative Raman spectra before and after the cracking (blue and red, respectively) of water-stored samples KOK04 (**top**) and CH01 (**bottom**), displaying the shift in Raman bands toward low frequencies after the cracking of water-stored samples.

## 4. Discussion

### 4.1. Stress Resistance of Opal

Cracking occurs when stress applied to a material exceeds what it can sustain. At the beginning of the 20th century, Griffith established a theory of fracture mechanics and created a criterion relating the energy released by a fracture at a certain loading level [64]. The critical stress, $\sigma_c$, that is necessary to fracture a material can be calculated with the following expression:

$$\sigma_c = \sqrt{\frac{2\gamma_s E}{\pi a}} \quad (1)$$

where $\gamma_s$ is the surface energy, $E$ is Young's modulus, and $a$ is the crack length inside the material. In this expression, $2\gamma_s$ defines the critical energy release rate, $G_c$. This theory applies to brittle materials. For plastic materials, a modification is required to account for the energy dissipation during plastic deformations. This energy is integrated in $G_c$ as

$$G_c = 2\gamma_s + G_{diss} \tag{2}$$

where $G_{diss}$ is the energy dissipated as unrecoverable inelastic deformation [65,66]. As surface and plastic dissipation energies are difficult to measure, other data are needed to estimate the critical stress. Based on Griffith's theory, Irwin [65] introduced the stress intensity factor (noted $K_{IC}$), which is related to $G_c$ by the following equation:

$$G_c = \frac{K_{IC}2}{E^*} \tag{3}$$

where for an isotropic material

$$E^* = \frac{E}{(1 - \nu^2)} \tag{4}$$

where $\nu$ is Poisson's ratio. The stress intensity factor has been measured on one natural opal by Simonton et al. [67] at 0.81 MPa·$\sqrt{m}$, consistent with the atomistic model and experimental data on amorphous silica (gel and glass [68–71]). Young's modulus of natural opals has been measured by Thomas et al. [72] between $49 \pm 2$ and $58 \pm 1$ GPa, which is lower but in the same order of magnitude as that measured for amorphous silica (several dozens of GPa depending on density [73–76]). Poisson's ratio has been estimated to be 0.2 for silica gel [77]. $G_c$, related to the surface energy by Equation (2), ranges from 9.56 to 12.86 J/m$^2$; thus, the surface energy, estimated from 4.78 to 6.43 J/m$^2$ by considering $G_{diss} = 0$, is consistent with what has been calculated or measured on amorphous silica with various degrees of hydration [71,78,79]. As the dissipation energy, $G_{diss}$, has not been assessed, we cannot know if the surface energy of opals is closer to that of a hydroxylated (hydrated) silica surface (around 1 J/m$^2$) or a non-hydroxylated (dry) silica surface (around 5 J/m$^2$) [79]. Considering the high state of the hydration of opal, we may suggest that an opal has a non-zero energy of dissipation.

With these few data available, we calculated the stress required for crack propagation according to the flaw size. Critical stress is approximately 400 MPa for a crack length of 1 µm, and it reaches approximately 13 GPa for a crack of 1 nm.

### 4.2. Cracking Processes

From the samples analyzed here, we differentiate two behavioral groups:

(i)   Cracking for air-stored samples, occurring at high temperatures (>250 °C) without any other effects;

(ii)   Cracking in water-stored samples, occurring at "low" temperature (<150 °C) and developing anisotropic features and peak shifts in the Raman spectra.

Anisotropy in a nominally isotropic material is generally attributed to the development of stress in the structure and is common in natural and artificial materials, including silica glass [80–84]. The shift in the Raman peaks' positions, being identical for non-cracking samples in the temperature range studied, could not be attributed to structural changes at the atomic level (or changes in the atomic structure) due to heating. In accordance with anisotropy, the Raman band's shift toward lower frequencies in stressed silica suggests the presence of tensile stresses [85,86]. Therefore, the cracking of water-stored samples is associated with stresses that remain active after structure fractures. By contrast, air-stored samples demonstrate no such residual stresses remaining in the opals' post-cracking. Given the observed difference in behavior both optically and spectroscopically, the physical origin of the cracking between the two sets of samples (air-stored and water-stored) must be different.

### 4.2.1. Decrepitation

Decrepitation is the formation of fractures resulting from the pressure induced by heating due to the thermal expansion of inclusions (often fluid) in their host mineral. The main condition for decrepitation to occur is that the fluid needs to be sealed (hermetic) or at least have negligible diffusivity [87]. The water diffusion coefficient in opals has been estimated to be negligible [88,89]. The TGA behavior of air-stored samples that were only observed to crack at elevated temperatures reveals that they do not release water until 300 °C (the first behavior described in Section 3.2). Thus, prior to heating, these samples retained water in the opal's structure. Some of these samples have been exposed to low pressure (1 mbar), and potential dehydration has been monitored by infrared spectroscopy as a function of time and pressure in a previous study [62]. All air-stored samples that showed cracking at elevated temperatures did not lose water as pressure was reduced, suggesting that water in air-stored samples is contained in a system of closed pores. Other air-stored samples that do not crack at high temperatures present TGA behaviors, which show water release once heating starts. This indicates that water is retained in a system of open pores; thus, the process of decrepitation does not occur.

As the above data suggest, water in the air-stored samples that crack at elevated temperatures is contained due to closed porosity. As the opal's formation is considered to take place in an aqueous solution near surface conditions [35,39,40], we may assume that the water-filled pores are full, without a gas phase. The expansion of water inside enclosed opal pores, therefore, provides the source of volume expansion observed in some opals up to a temperature at which dehydration starts (also near 300 °C) [48,90,91]. We thus propose that air-stored samples were subjected to decrepitation-induced cracking by the thermal treatment.

Pores that are hermetically filled with water may be defined as fluid inclusions but at a nanometer- to micrometer-sized range that is invisible to the naked eye. Each pore is thus a non-extensible volume, similarly to the common fluid inclusions studied to trace the geological conditions needed for mineral formation [87,92,93]. The isochoric relation (the evolution of pressure with temperature at a constant volume) significantly varies with the composition of the fluid and its density. Considering the formation of opals, we may assume that water is the main fluid that is entrapped. The only fluid inclusions that have been studied were found in Mexican opals. These fluid inclusions contained a water-rich fluid, with sometimes small NaCl and $CO_2$ contents [37]. As most gem opals form in near-surface conditions and aqueous fluids (especially for Australian and Ethiopian opals [35,39,40]), we may assume that the trapped fluid densities are close to that of liquid water. This assumption makes it possible to calculate the pressure inside an inclusion at a given temperature. Even if air-stored samples have the required properties for decrepitation, is the expansion of water sufficient to crack the opal's structure? Most of the samples were observed to crack upon heating to 300 °C, where isochoric relations (for a density near 0.95 if we assume that water contained dissolved ions) produced pressure from 200 to 600 MPa [87,92,94]. This estimation range is large and is mainly due to the lack of data on the exact density and chemical composition of opal's fluid inclusions, both having a significant impact on pressure value.

We note that the induced pressure, near 300 °C, is of the same order of magnitude than the stress required to create cracks in an opal. By contrast, at less than 150 °C, where water-stored opals crack, the pressure is only a few dozen of MPa. Therefore, decrepitation probably cannot explain the cracking of water-stored samples, considering the temperature at which it occurs.

Combining our data with existing theories, we suggest that samples stored for many years in air had time to release water due to their open porosity and should not experience cracking unless subjected to significant heating. In nature, opal is rarely exposed to such temperatures without significant a change in structure, as observed in diagenetic sequences where opal transforms into quartz with time and temperature [95–97]. However, elevated temperatures could be reached during the fashioning of a stone (via friction),

especially during pre-forming. We propose that, without heating, air-stored samples should remain stable.

### 4.2.2. Drying Shrinkage

Drying shrinkage is the decrease in the volume of a porous structure and the concomitant decrease in porosity during a loss of water. This phenomenon has been extensively studied in materials engineering, especially on silica gel. Silica gel is commonly synthetized by super-saturating a solution in silica by mixing TEOS (tetra-ethyl-orthosilicate), water, and ethanol [15–17]. The polymerization of the gel (sometimes called the aging of the gel) occurs in conditions where fluids fill all the pores of the gel. Drying aims to remove fluids, and two main steps have been identified: a constant rate period (CRP) and a falling rate period (FRP) [98–102]. CRP occurs initially with a constant evaporation rate, and the structure of the gel shrinks due to the capillary stresses exerted by the liquid inside the poreosity During this step, the gel is compliant enough to shrink without cracking, and evaporation occurs at the surface of the gel. In cases where the gel remains compliant throughout the drying process, a dry xerogel is produced, and porosity disappears. Under these conditions, the pore fluid contained in the gel flows from the interior to the surface and evaporates; thus, no liquid–vapor interface exists inside the gel. The aging of the gel continues during this period by creating new Si-O-Si bonds until a limiting volume for densification is reached, and the gel retains its pore structure. At this critical point (called the critical point), the liquid–vapor interface enters the gel. The capillary stress is at a maximum, and it may be great enough to initiate (and propagate) cracks. After the critical point, the evaporation rate decreases, and the gel drying enters the FRP, during which the meniscus of the liquid–vapor interface moves inside the gel's structure.

Using the theory developed for silica gels, we may calculate the pressure induced by the capillary forces on the gel's structure. This pressure is related to the contact angle, $\theta$ (a measurement of the hydrophilicity between the solid and the fluid), the liquid–vapor interfacial energy, $\gamma_{lv}$, and the pore's size, $r_p$ [99,103]. The maximum pressure can be approximated by the following equation.

$$P_{max} = \frac{2\gamma_{lv}\cos\theta}{r_p} \tag{5}$$

The pressure induced, varying according to these parameters, could reach several hundreds of MPa [98,104,105], which is the same order of magnitude as the opal fracture strength. The maximum capillary pressure is the pressure induced by liquid–solid interfaces (i.e., at the pore wall) at the critical point. A uniform pressure does not induce stress, as stress is a result of a pressure difference. The total stress during the CRP, derived from Darcy's law and the mechanical response of the gel, may be approximated for the surface of a plate [99]:

$$\sigma = \left(\frac{1-2\nu}{1-\nu}\right)\left(\frac{L\eta_l\dot{V}_E}{3D}\right) \tag{6}$$

where $\nu$ is the Poisson's ratio of the material, $L$ is the half-thickness of the plate studied, $\eta_l$ is the viscosity of the liquid, $V_E$ is the evaporation rate, and $D$ is the permeability of the sample [98,99,104]. This expression is more complex at the critical point and incorporates the maximum capillary pressure seen above. Many parameters, such as evaporation rate and permeability, remain elusive for the application of this equation in opals. However, the fact that the maximum capillary pressure reaches values consistent with the opal's fracture strength demonstrates the relevance of this process in explaining cracking.

Drying shrinkage explains many of the observations:

- It involves a tensile stress that continues after the crack has released part of the stress (as evaporation continues and sometimes accelerates [98,99,104]), as observed in water-stored opals that have cracked via the observed anisotropy and the shift in Raman bands.

- Earlier studies have suggested that a low evaporation rate partly avoids cracking [4,5], as suggested by Equation (6): if $V_E$ tends to 0, so does the stress.
- The tension induced by the fluid interface tends to increase the stress at the drying front; thus, as evaporation is initiated at the surface, the surface tends to contract faster than the interior, producing a cracked surface, as is commonly observed [1,63]. We could assume that if the drying front is already inside the sample but the evaporation rate rapidly increases, such as when opal is initially extracted, the stress strongly increases. This may result in a cracked volume inside a transparent opal, with a shape related to the outside shape of the opal piece and, hence, to the shape of the drying front [1].

*4.3. Practical Considerations*

We have identified and presented two processes, decrepitation and drying shrinkage, that satisfactorily explain our results and many earlier observations about opal cracking. However, the driving process differs according to the drying state of the sample. A natural opal that has open pores (interconnected pores open to the atmosphere) that contain water should crack due to drying shrinkage, whereas an opal that is already dry is subjected to decrepitation only if heating is sufficient. Differentiating each opal type could be achieved by measuring whether or not the opal can still lose water, but it should be noted that a drying opal does not necessarily crack; this condition is essential but not always sufficient to induce cracking. We may assume that all opals, immediately after extraction, experience a period of dehydration to equilibrate with the atmosphere. At the mine site, it is useful to expose freshly mined opals to natural dehydration for some time (months, for example) to determine which opals will start to "craze" in some way and which are stable and may be further fashioned and put on the market.

The drying of engineered silica gel has been studied to manufacture large blocks without cracks. Several processes of drying are used to avoid cracks: by limiting the capillary pressure (Equation (5)), by reducing the interfacial energy, or by increasing the pores' radii. As natural gem opals are already formed, one cannot change its pore radius, but we can intervene by reducing the interfacial energy to zero (and thus the capillary pressure) by using supercritical drying, a process commonly used to prepare aerogels [98,103,106–109], which removes molecular water from natural opals [63]. Another method is freeze drying, where silica gel is first frozen and dried under a vacuum [103,110,111], a process that has yet to be applied to natural opals. The removal of molecular water prevents drying shrinkage, but it has been demonstrated in gels that later rehydration would reset the process and could induce cracking at a later stage [112,113]. In the treatment proposed by Filin and Puzynin [63], after supercritical drying, the pores are filled by silica to avoid any rehydration.

Preserving opals in water to avoid cracking is explained by the drying shrinkage theory; in water, the evaporation rate is null. However, once an opal is exposed to the atmosphere, drying starts, which may initiate cracking. We may also assume that water could alter the silica framework by dissolving silica until the pore solution is saturated, thus lowering the mechanical resistance. Hence, changing the preserving water may weaken the sample. However, considering the low solubility of amorphous silica in water, circa 100 ppm at ambient temperatures and neutral pH [96,114,115], this effect should be marginal. The influence of the evaporation rate on drying stress also explains some observations reported by opal dealers who observed that opals preserved in a safe do not evolve and start to crack once they are removed. The evaporation rate in a hermetic box decreases until it reaches zero when the air is saturated by water. Once the safe is open, the atmosphere is renewed, and the evaporation rate increases, restarting the drying process. The common practice of placing a glass of water close to opals while they are kept in a showcase or a hermetic place will decrease the evaporation rate and, thus, partly avoid cracking. Samples that have significant permeability, such as hydrophane opals, exhibit lower drying stress, as expressed in Equation (6). This explains the common observation

that hydrophane opals are rarely subject to cracking. Moreover, based on Equation (6), drying shrinkage is not related to the water content in opals but to the pore's structure.

Establishing a theoretical framework based on drying-shrinkage-induced cracking opens a path toward predicting if an opal is prone to cracking. In order to develop a method, we first need to assess the fracture strength of natural opals. Reported mechanical properties of opals are scarce in the literature, and they have been measured only for synthetic and Australian opals [67,72]. For silica xerogel, its mechanical properties have been related to density [76,116]. If natural opals share this property, density measurements could be a simple and non-destructive way of assessing the mechanical properties of a sample. Secondly, it is necessary to estimate the drying stress required to initiate cracking. To achieve this, the measurement of the pore size and the permeability of a sample is required. Even if these properties could be measured by non-destructive methods (e.g., differential scanning calorimetry or infrared spectroscopy [52,58,62]), work is required to adapt these techniques for the gem market.

## 5. Conclusions

The present study identified two main processes involved in opal cracking: decrepitation and drying shrinkage. Our experiments suggest that there are two different categories of cracking opals: opals that crack only during heating to high temperatures (>250 °C) and opals that crack even at ambient conditions. Both opal categories differ by their storage history: The opals in the first category were allowed to dry at ambient conditions (temperature, pressure, and humidity) for an extended period of time, whereas the opals in the second category have been stored in water (thus not drying) since their extraction. Decrepitation explains the reaction of the first category to heating, and drying shrinkage explains cracking in the second category. The theories of each process also confirm the practical observations collected from the gem market and help isolate the driving parameters that could be instrumental to creating an analytical procedure for predictions and thus help prevent the cracking of gem opals.

**Supplementary Materials:** The following supporting information can be downloaded at: https://www.mdpi.com/article/10.3390/min13030356/s1.

**Author Contributions:** Conceptualization, B.C.; methodology, B.C., V.M., F.G., B.R. and P.S.T.; validation, B.C. and B.R.; formal analysis, V.M., F.G., B.C. and P.S.T.; investigation, B.C., V.M. and F.G.; resources, B.R., E.F. and P.S.T.; data curation, V.M., F.G., B.C., B.R. and P.S.T.; writing—original draft preparation, B.C.; writing—review and editing, all authors; visualization, B.C.; supervision, B.C. and B.R.; project administration, B.C. and B.R.; funding acquisition, B.C. All authors have read and agreed to the published version of the manuscript.

**Funding:** This research was partly funded by the Eduard Gübelin Association for Research & Identification of Precious Stones.

**Data Availability Statement:** All data on water content from the literature are available in Supplementary Materials. All others data acquired are available upon request.

**Acknowledgments:** Parts of the analyses were possible thanks to the Eduard Gübelin Research Scholarship 2016 sponsored by the Eduard Gubelin Association for Research & Identification of Precious Stones. The authors would like to thank Selamawit Dagnachew, who collected the water-stored samples in the Wegel Tena mines, Erwan le Menn and his assistance during the experiments, and Laurent Lenta for the preparation of the samples.

**Conflicts of Interest:** The authors declare no conflict of interest. The funders had no role in the design of the study; in the collection, analyses, or interpretation of data; in the writing of the manuscript; or in the decision to publish the results.

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
