# Peer review of "Cracking of Gem Opals"

_minerals, doi:10.3390/min13030356_

Round 1

Reviewer 1 Report

Except for minor approximative vocabulary issues, the paper is of really good quality, with a subject that directly concerns and is applicable to the trade market of gems. The study will serve not only the miners, the dealers, but also the end-buyers, as it provides keys to understand the nature of opals and why it has the bad reputations of being unstable. It also provides insights on common, but not understood knowledge of why and how an opal should be preserved to prevent cracking. The research provides experiments related to the subject matter in order to understand the physical properties involved in opal cracking.

The authors will find my comments in the attached document.

My main comment : Please remove the use of "old" and "fresh" opals as they are confusing and misleading.

Reviewer 2 Report

The article is fascinating and provides physicochemical elements that help to understand the opal cracking process. The content, structure, results and conclusions are adequate for the publication of this manuscript in Minerals. On the other hand, I recommend improving the language to arrive at a more obvious understanding of certain sentences and to upset many typos in the entire document.

Reviewer 3 Report

Review: Cracking of gem opals

Summary

This manuscript reports the formation of opal cracking using visual examinations, thermogravimetric analysis and Raman spectroscopy. The opal samples divided in two groups opal samples that stored in water after extraction, and others are stored in air for more than a year, Based on the several observations, for the old samples, cracking was ascribed to super-heated water induced decrepitation, and the cracking in the fresh samples was related to drying shrinkage which correlates with the anecdotal reports from the gem trade. Overall, the ms is well written and can be published at Minerals with minor revision. 

Some detailed comments

L49: What is opal? This is a colloquialism; the readers should know this.

L60-65: this part does not like a scientific expression, need combination.

L66-67: X-ray diffraction, Raman spectroscopy and infrared spectroscopy is important to investigate minerals within hydration. Thus, this part can be expanded to other minerals that including hydration (e.g., apatite), not only opal, making statement more convincing. you can cite: Xu, B.; et.al, Apatite halogens and Sr–O and zircon Hf–O isotopes: recycled volatiles in Jurassic porphyry ore systems in southern Tibet. Chem. Geol. 2022, 120924.

Figure.2 is not clear, it would be good if you can expand a larger one.

L195-206: this need combination.

L378-391: this need combination again.
